# SPURIOUS FEATURES IN CONTINUAL LEARNING

## ABSTRACT

Continual Learning (CL) is the research field addressing learning without forgetting when the data distribution is not static. This paper studies spurious features' influence on continual learning algorithms. We show that continual learning algorithms solve tasks by selecting features that are not generalizable. Our experiments highlight that continual learning algorithms face two related problems: (1) spurious features (SF) and (2) local spurious features (LSF). The first one is due to a covariate shift between training and testing data, while the second is due to the limited access to data at each training step. We study (1) through a consistent set of continual learning experiments varying spurious correlation amount and data distribution support. We show that (2) is a major cause of performance decrease in continual learning along with catastrophic forgetting (CF). This paper presents a different way of understanding performance decrease in continual learning by highlighting the influence of (local) spurious features in algorithms performance.

## 1 INTRODUCTION

Feature selection is a standard machine learning problem. Its objective is to improve the prediction performance, provide faster and more effective predictors, and provide a better understanding of the underlying process that generated the data (Guyon & Elisseeff, 2003). In this paper, we are interested in improving prediction performance in the presence of spurious features. Spurious features arise when features correlate well with labels in training data but not in test data. Learning algorithms that rely on spurious features will generalize badly to test data.

In continual learning (CL), the training data distribution changes through time. Hence, we could expect that spurious features (SFs) in one time-step of the data distribution will not last. A continual learning algorithm relying on a spurious feature to solve a task can then be resilient and learn better features later, given more data. Algorithms can aim to detect and ignore spurious features learned in the past (Javed et al., 2020). An example of a task with spurious features could be a classification task between cars and bikes, but in the training data, all cars are red, and all bikes are white, but it test data, both are in a unique blue not available in train data. A model could easily overfit the color to solve the task while it is not discriminative in the test data. This problem is notably caused by a covariate shift between train and test data. In CL, we would expect future tasks to bring pictures of cars and bikes of other colors to learn better features.

On the other hand, in CL, the second type of spurious feature can be described: *local spurious features*. Local features denote features that correlate well with labels within a task (a state of the data distribution) but not in the full scenario. In opposite to the usual *spurious features*, this problem is provoked by the unavailability of all data, for example, because only red cars and white bikes are currently available in train data, but in the test data, there are **also** cars and bikes of colors not seen yet or seen in the past. Overall, there is no significant covariate shift between all train data and test data. It is, therefore, a problem specific to continual learning.

This paper investigates both the problem of spurious features (with covariate shift) and local spurious features (without covariate shift) in CL as shown in Fig. 1. Our contributions are: (1) We propose a methodology to highlight the problems of spurious features and local spurious features in continual learning. (2) We create a binary CIFAR10 scenario *SpuriousCIFAR2* inspired by colored MNIST to experiment with spurious correlations. (3) We propose a modified version of Out-of-Distribution (OOD) generalization methods for continual learning and evaluate them on *SpuriousCIFAR2*. (4) We identify local spurious features as a core challenge for continual learning algorithms along with CF.

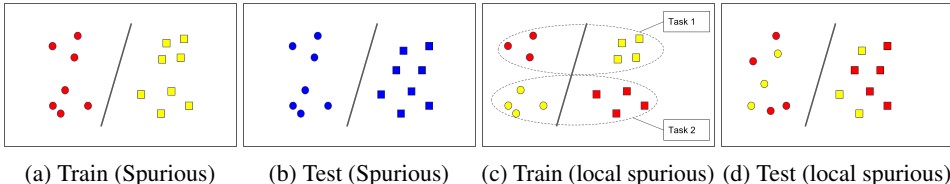

(a) Train (Spurious)     (b) Test (Spurious)     (c) Train (local spurious)  (d) Test (local spurious)

Figure 1: **Spurious features and local spurious features.** If the task is to distinguish the squares from the circles. In Fig. 1a and 1b, the color is a spurious feature because there is a covariate shift between train and test data. In Fig. 1c and 1d, we observe two tasks of a domain-incremental scenario, the colors are locally spurious in tasks 1 and 2. Even if there is no significant covariate shift between train and test full data distribution, colors appear discriminative while looking at data within a task.

We expect this paper to improve the understanding of the continual learning problem.

## 2 RELATED WORK

In large part of the continual learning bibliography, algorithms assume that to avoid catastrophic forgetting (CF), they should not increase the loss on past tasks (Kirkpatrick et al., 2017; Ritter et al., 2018). It leads to the definition of interference/forgetting of (Riemer et al., 2019): $\frac{\partial L(x_i, y_i)}{\partial \theta}$ . $\frac{\partial L(x_j, y_j)}{\partial \theta} < 0$, $\forall (x_i, y_i) \in \mathcal{T}_i$ and $\forall (x_j, y_j) \in \mathcal{T}_j$ with $j > i$, $< \cdot >$ is the dot product operator. Following this definition, increasing the loss on past tasks necessarily leads to a performance decrease. However, the algorithm might have learned spurious/local features that need to be forgotten to improve performance. Hence, the loss needs to be temporarily increased to reach a more general solution, and optimizing the interference equation could be counterproductive. On the same line, the presence of spurious features could be adversarial to most continual regularization strategies. For example, if we measure weights importance with Fisher information, high importance will be given to weights using spurious features, and regularization will penalize their modification. Regularization could protect features that generalize poorly to the test set.

Vanilla rehearsal or generative replay is a good solution to not forget meaningful information and to deal with spurious features. By replaying old data, algorithms simulate an independent and identical distribution (iid) and avoid local spurious feature problems. Replay methods have been shown in the bibliography to be efficient and versatile even in their most straightforward form (Prabhu et al., 2020). Notably, CL state-of-the-art on ImageNet uses replay (Douillard et al., 2020; Zhao et al., 2020).

The research field that usually deals with spurious correlations is the out-of-distribution (OOD) generalization field. This field has received a lot of attention in recent years, especially since the Invariant Risk Minimization (IRM) (Arjovsky et al., 2019) paper. OOD approaches target training scenarios where there are several training environments within which different spurious features correlate with labels. The goal then is to learn invariant features among all environments to build an invariant predictor in all training environments and potentially any other (Arjovsky et al., 2019; Ahuja et al., 2021; Sagawa et al., 2019; Pezeshki et al., 2020). This paper will adapt some of those approaches for continual learning to evaluate how those approaches can deal with sequences of tasks.

## 3 PROBLEM FORMULATION

This section introduces the spurious features problems in a sequence of tasks. The goal is to present the key types of features, namely: general, local, and spurious features.

**General Formalism:** We consider a continual scenario of classification tasks. We study a function $f_\theta(\cdot)$, implemented as a neural network, parameterized by a vector of parameters $\theta \in \mathbb{R}^p$ (where p is is the number of parameters) representing the set of weight matrices and bias vectors of a deep network. In continual learning, the goal is to find a solution $\theta^*$ by minimizing a loss $L$ on a stream of data formalized as a sequence of tasks $[\mathcal{T}_0, \mathcal{T}_1, ..., \mathcal{T}_{T-1}]$, such that $\forall (x_t, y_t) \sim \mathcal{T}_t$ $(t \in [0, T-1])$, $f_{\theta^*}(x) = y$. We do not use the task index for inferences (i.e. single head setting).

To describe the different types of features, let $z$ be a feature and $x \sim \mathcal{D}$ a datum point in dataset $\mathcal{D}$. We define $w(.)$ a function which returns 1 if $z$ is in $x$ and 0 if not. $w(.)$'s output is binary for simplicity. Then, for all data with a label $y$ in the dataset $\mathcal{D}$, we can compute the correlation $c(\mathcal{D}, z, y) = correlation(w(z, x) = 1, Y = y)$, which estimates how a feature correlates with the data of a given class. We can then define discriminative features as:

Table 1: Summary of characteristics of the types of features. For a feature $z$ of a class $c$, we denote if it verify (1) on different data setting, a single task $\mathcal{T}_t$, the whole scenario $\mathcal{C}_T$, the test set $\mathcal{D}_{te}$.

| Name | $\mathcal{T}_t$ | $\mathcal{C}_T$ | $\mathcal{D}_{te}$ |
|---|---|---|---|
| Good Feature ($z_+$) | ✓ | ✓ | ✓ |
| Spurious Feature ($z_{spur}$) | ✓ | ✓ | ✗ |
| Local Feature ($z_{loc}$) | ✓ | ? | ? |
| Local Spurious Feature ($z_{spur;t}$) | ✓ | ✗ | ✗ |

$z$ is discriminative for class $y$ in $\mathcal{D}$ if:

$$\forall y' \in \mathcal{Y}, y \neq y' \quad c(\mathcal{D}, z, y) \gg c(\mathcal{D}, z, y') \tag{1}$$

$\mathcal{Y}$ is the set of classes in $\mathcal{D}$. In other words, $z$ is discriminative for $y$ if it correlates significantly more to $y$'s data than to the data of any other class. Then a good feature $z_+$ for a class $y$ respects (1) for training data $\mathcal{D}_{tr}$ and test data $\mathcal{D}_{te}$.

**Spurious Features (SF) and Local Spurious Features (LSF):**

A spurious feature $z_{spur}$ for a class $y$ respects (1) for training data $\mathcal{D}_{tr}$ but not for test data $\mathcal{D}_{te}$. A spurious feature is well correlated with labels in training data but not with testing data. Hence, learning from $z_{spur}$ may offer a low training error but high test error. The presence of $z_{spur}$ is due to a covariate shift between train and test distribution which changes the feature distribution.

In continual learning, the covariate shift between train and test $z_{spur}$ may also lead to poor generalization. Further, the features can be locally spurious, e.g., they correlate well with labels within a task but not within the whole scenario. We name them *local spurious features* (LSF). We illustrate the difference between spurious features and local spurious features in Figure 1.

At task $t$, A local spurious feature $z_{spur;t}$ respects (1) for a class $y_t$ in task $\mathcal{T}_t$, but not for the whole scenario $\mathcal{C}_T$. $z$ is a LSF for a class $y$ in $\mathcal{T}_t \sim \mathcal{C}_T$, with $t \in [\![0, T-1]\!]$:

$$\begin{aligned} \text{if } \forall\, y' \in \mathcal{Y}_t, y \neq y' \quad & c(\mathcal{T}_t, z, y) \gg c(\mathcal{T}_t, z, y') \\ \text{and } \exists\, y'' \in \mathcal{Y}, y \neq y'' \quad & c(\mathcal{C}_T, z, y) \not\gg c(\mathcal{C}_T, z, y'') \end{aligned} \tag{2}$$

$\mathcal{Y}_t$ is the classes set in task $\mathcal{T}_t$ and $\mathcal{Y}$ is the classes set in the full scenario $\mathcal{C}_T$ composed of $T$ tasks.

A LSF $z_{spur;t}$ correlates well with a label on the current task but not on the whole scenario. $z_{spur;t}$ can be extended from a single task $\mathcal{T}_t$ to all task seen so far $\mathcal{T}_{0:t}$ without loss of generality.

**Global vs Local Solution:** We assume that machine learning models solve tasks by learning to detect (or select) features that correlate well with labels. Then, while learning on a task $t$, we can distinguish a local solution $\theta_t^*$, satisfying for the current task $\mathcal{T}_t$, from a global solution $\theta_{0:T}^*$ that is satisfying for whole scenario $\mathcal{C}_T$ (past, current, and future tasks).

Similarly, we can differentiate local and global features, contributing to local and global solutions. The global features are the good features $z_+$ that are useful for the solution of the scenario. Unfortunately, at time $t$, we can not know if a feature is part of $z_+$ without access to the future. Therefore, algorithms should learn with their current data but update their knowledge afterwards, given new data. For example, in classification, the discriminative features for a given class depend on all the classes. Therefore, when new classes arrive, discriminative features can become outdated in class-incremental scenarios.

## 4 INFLUENCE OF SPURIOUS FEATURES (SFs) ON CONTINUAL LEARNING

This experimental section studies how continual learning algorithms can deal with spurious features. We design a scenario with spurious features that change at each task. We create a set of scenarios with gradual correlations between spurious features and labels. We evaluate various baselines to assess continual learning capabilities (Sec. 4.3.2) in such scenarios. We also experiments with potential solutions to deal with spurious correlation.

### 4.1 SETTING

We propose a benchmark "SpuriousCIFAR2" inspired by colored MNIST (Arjovsky et al., 2019) with CIFAR10 but without label noise. We convert the ten-way classification dataset into a binary dataset. The new classes are "transportation means versus not transportation means", i.e., cars, trucks, ships, airplanes, and horses versus the other classes: birds, cats, dogs, deers, frogs. The goal is to have a simple setting more challenging than colored MNIST with features that are built upon all three color channels.

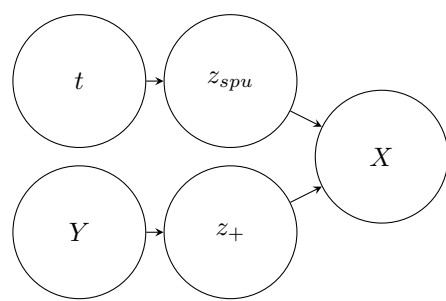

The spurious feature is a square of color. We sample two colors randomly (one per class) and add a $2 \times 2$ pixels square randomly positioned in the images (Ap. E). We can vary the percentage of images with the spurious feature to reduce or grow the correlation between the spurious feature and the labels. A correlation of 1 means that all images have a colored square.

Figure 2: Data generation model of SpuriousCIFAR2 scenario. The good features $z_+$ are generated from the labels and $z_{spu}$ are generated from the task label.

The scenario is then a sequence of SpuriousCIFAR2 datasets with different colors of spurious features. The test set of each scenario is the binarized version of CIFAR10 test set without any spurious features. We created the sequence of tasks using the *Continuum* library (Douillard & Lesort, 2021). We illustrate two environments and the test set in Sec. E, and the data generation process in Fig. 2.

**Setting Goal:** This setting is made to highlight how spurious features can disrupt CL algorithms and discuss the problems that spurious correlation can bring to existing approaches. Moreover, it evaluates the capacity of algorithms to modify their past knowledge to improve test accuracy.

## 4.2 APPROACHES

First, we experiment with a classical vanilla replay (rehearsal) method and simple finetuning. Finetuning baseline consists of training with Adam optimizer without any CL mechanism. For rehearsal, the replay buffer is constructed by randomly selecting $N$ samples per class. The buffer is then sampled to keep class distribution uniform over all classes seen so far. Balancing samples distribution over classes is made to avoid the challenge of training on imbalanced datasets (cf Sec. I in the appendix for details).

For the existing OOD approaches, we compare continual versions of IRM (Arjovsky et al., 2019) and the state-of-the-art OOD classification methods IB-ERM, IB-IRM (Ahuja et al., 2021), GroupDRO (Sagawa et al., 2019) and Spectral Decoupling (Pezeshki et al., 2020). OOD approaches are algorithms designed to be trained on multiple environments in a multi-task fashion. IRM (Arjovsky et al., 2019) (invariant risk minimization) is an approach that uses multiple environments to learn invariant features and improves empiric risk minimization (ERM) generalization. IB-ERM augments this approach with a regularization terms based on an **i**nformation **b**ottleneck constraint. GroupDRO (group Distributionally Robust Optimization) learns models that minimize the worst-case training loss over the set of environments. (Sagawa et al., 2019) proposes to couple group DRO models with a strong regularization term such as L2 regularization or early stopping to improve generalization. Spectral Decoupling (Pezeshki et al., 2020) proposes a regularization term that maximizes the number of features an algorithm learns to avoid relying only on spurious features. The continual version of OOD approaches simulates multiple environments by replaying data of past tasks. The adaptation of all those methods is then to add a replay buffer to the algorithms to train continually the baseline through the sequence of tasks. The replay buffer simulates the growing number of environments for the OOD approaches. In such context, replay (rehearsal) is equivalent to the ERM (empiric risk minimization) baseline in OOD literature. We choose empirically a replay buffer storing 100 samples per class for both OOD approaches and vanilla replay (cf appendix F for HPs selection).

## 4.3 EXPERIMENTS

### 4.3.1 PROBLEM HIGHLIGHTS

**Overfitting the spurious features:** To assess that algorithms overfit on the spurious features, we train the model on a single task with spurious features. We compare the test accuracy on data without

spurious features (final test set) with the test accuracy on data with spurious features (evaluation set). If the test accuracy is good with spurious features and bad without, the algorithm overfits the spurious features. Fig. 3, show exactly this phenomenon, test accuracy on task 0 is near-perfect accuracy, while on the final test set, the accuracy is near-random prediction. This figure shows that the artificial spurious features cause the expected learning behaviors: the model overfits and generalizes poorly.

**Instability:** In Fig. 4, we assess the test accuracy $A$ at each epoch over the whole sequence of 10 tasks. This figure indicates two interesting pieces of information. First, even in the 100% spurious correlation, i.e. all images have a square of colors, baseline models can learn at some point a good solution. Secondly, even when they learn a good solution, they are very unstable and can easily forget a good solution. To lighten the influence of instability on the evaluation metrics, in the later experiments of this section, we report the average test accuracy after each task, which we note $\Omega = \frac{1}{T} \sum_{i=0}^{T-1} A_i$ instead of reporting the final test accuracy $A_{T-1}$.

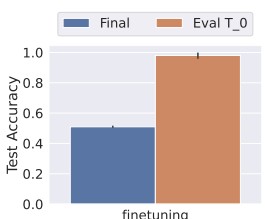

Figure 3: Overfitting the spurious features on a single task setting.

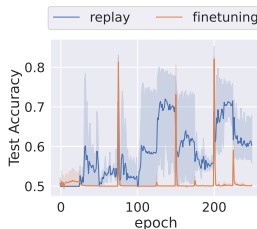

**Comparing CL Baselines with OOD Baselines:** We now assess how baselines adapted from the OOD field behave in the 100% spurious correlation setting. In our experiments, *IB-ERM*, *SpectralDecoupling* and *GroupDRo* show some interesting improvement over rehearsal and finetuning baselines ( Fig. 5). On the other hand, *IRM* and *IB-IRM* performed poorly.

Those experiments show us that we can have some improvement over replay and finetuning baselines when there is 100% correlation between spurious features and labels. Note that there is much variance in results because of the instability mentioned earlier. Moreover, the improvements stay far from a satisfying accuracy. Indeed, the average performance of the baselines on SpuriousCIFAR2 is below 70% of average accuracy. Meanwhile, we trained a model on the CIFAR2 dataset without spurious features and reached 96.73% of accuracy. The next experiments will analyze performance while gradually growing the spurious correlations.

Figure 4: Generalization and forgetting: test accuracy in a sequence of tasks (25 epochs per task).

### 4.3.2 INFLUENCE OF SPURIOUS CORRELATION

In this section, we aim to answer the question "how does the level of spurious correlation influence learning algorithms?". From a stream of tasks, we can expect that the 100% spurious correlation is quite rare. Hence, we will investigate the setting of lower spurious correlation in this set of experiments. We study the 25%, 50% and 75% spurious correlation cases along the 100% correlation.

Fig. 6 show that lowering the spurious correlation makes rehearsal and finetuning (baseline) the best approaches. Those results indicate that the OOD baselines are most interesting in the very high spurious correlation setting but are not very interesting when the spurious correlation is lower or equal to 75%. It might seem counter-intuitive that finetuning is over the best baseline in a continual learning setting. Still, only the spurious correlation change in our scenario, so the global features needed to solve all tasks are in each task. It is then reasonable that with a lower spurious correlation finetuning works well.

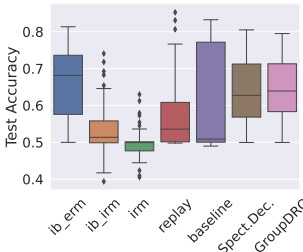

Figure 5: Accuracy with local 100% correlation between features and labels

The scenario could be made harder by only keeping a subset of the dataset in each task, i.e., lowering the support of the data distribution (cf in appendix B). However, in the experiments, we can note that lowering the support makes the task harder because there are fewer data per task and removes the full observability of data. It means that good/global features are no longer always observable and that some good features can be available in some tasks but not others. Nevertheless, our results on the support showed that it has a low influence on the results. This is probably because replay makes everything observable at the last tasks and overcomes partial support challenges. It is similar

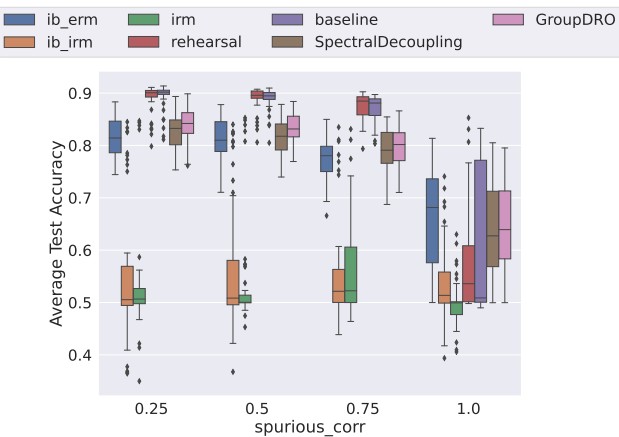

Figure 6: Averaged accuracy $\Omega$ on 10 tasks over various amount of spurious correlation between spurious features and labels.

to creating several environments with separate splits of the full distribution as in DomainBed datasets (Gulrajani & Lopez-Paz, 2020).

### 4.3.3 POTENTIAL SOLUTIONS TO LOWER IMPACT OF SPURIOUS FEATURES

A solution to prevent models from relying too much on spurious features is (1) to force them to learn/use more features or (2) to try to ignore the spurious features. We experiment with (1) by using a regularization strategy to maximize the number of features selected. This regularization is different from regularization methods designed to not forget. We experiment with (2) by using a model pre-trained on a trusted task. The idea of the pre-trained model is that it will ignore noisy features, such as SFs.

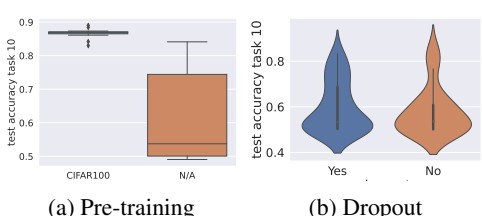

(a) Pre-training        (b) Dropout

Figure 7: Canceling noisy spurious features with pre-trained models.

**Using a pre-trained model:** We can use a pre-trained model on a trusted data source to ignore spurious features. For example, the spurious feature in our setting is noise, so if we use a pre-trained model on a known dataset such as CIFAR100, the model filter out noisy features and can significantly improve results. This approach, experimented in Fig. 7a, shows clearly that using a pre-trained model can erase the problem of noisy spurious correlation. This solution is convenient, but it assumes we have a compatible trusted set of data (or a trusted model) and that the spurious features are noisy. More details on noisy features in appendix C.

**Maximizing the amount of features selected:** If we can not use the previous solution, another potential solution is to learn to select as many features as possible that could help solve the problem. A famous solution to maximize the features learned is **dropout**. Dropout randomly replaces some activations value with a zero for inference to force the model to learn more robust features. It has notably been widely experimented in continual learning (Goodfellow et al., 2013; Mirzadeh et al., 2020). We experimented 0.25, 0.5, and 0.75 amount of dropout just before the last linear layer. However, in our experiments, it did not show any improvement (cf. Fig. 7b).

Nevertheless, on a similar idea as dropout, the **spectral decoupling** approach is designed to address the gradient starvation problem. The gradient starvation problem arises when the loss is minimized by capturing only a subset of features relevant to the task, despite the presence of other predictive features that fail to be discovered (Pezeshki et al., 2020). Spectral decoupling is designed to discover supplementary features even with minimal train error. As dropout, it enables the possibility to learn additional features that could help to improve the test error. The experiment in section 4.3.1 illustrated in Fig. 5 indeed shows that in the 100% spurious correlation experiment, this strategy greatly improves simple rehearsal proving the potential of the idea.

**Conclusion** We proposed some trivial solutions to illustrate how supplementary knowledge or assumptions on the spurious features might help prevent or fix bad learning behavior. However, it would probably not work as easily in a setting with more complex spurious features.

Experiments in sections 4.3.2 and 4.3.3 investigate how algorithms learning continually can deal with spurious features. We created a benchmark with spurious features and empirically investigated algorithms' performance when varying the correlation of spurious features with labels. In the next section, we will investigate local spurious features, which are a type of spurious feature specific to CL. We will investigate if those features may cause a performance decrease in CL algorithms. We shown that models can overfits spurious features easily, in the next section, we investigate if local spurious features (LSF) can exist in exisiting original datasets, and if they can be easily be overfit and influence models performance.

# 5 INFLUCENCE OF LOCAL SPURIOUS FEATURES (LSF) IN CONTINUAL LEARNING

In the previous section, we study how spurious features impact continual learning algorithms. In this section, we show that local spurious features (LSFs) may also lead to a performance decrease, even in scenarios composed of usual data.

## 5.1 LOCAL SPURIOUS FEATURES SETTING

Our goal is to investigate if some LSFs exist in continual learning scenarios without being manually added. To demonstrate their existence, we will show that they influence the results. Their influence is incarnated by a model that learns independent tasks correctly, but that can not generalize to tasks together.

**Data:** We experiment with classical datasets without modification to create class-incremental scenarios as in most continual learning literature. We use CIFAR10, OxfordPet, OxfordFlowers and CUB200 datasets with pretrained models: a resnet model on CIFAR100 for CIFAR10 and for the other datasets we use VGG, Resnet, Alexnet and googlenet pretrained on Imagenet from torchvision library. We create the scenarios by splitting each dataset into 5 tasks with disjoint sets of classes.

**Model:** We use frozen pretrained models: a resnet model on CIFAR100 for CIFAR10 and for the other datasets we use VGG, Resnet, Alexnet and googlenet pretrained on Imagenet from torchvision library. We assume that the pre-trained models provide a feature space sufficient for the continual downstream scenarios. They are used frozen with a linear classifier on top. We also assume that algorithms select features that correlate well with labels to solve the current task. **The goal is then to show that classifiers rely on local spurious features while learning continually.** If it is the case, this will lead to a good performance on the tasks but not in the full scenario. The training is realized with a simple finetuning without any replay or regularization.

**Approach:** We train a linear classifier per task (multi-head). Hence, while training, inference considers one head at a time for classification. The softmax function is applied only to the classes of a single head. We note the test performance in *multi-head* $A_{te-local\_softmax}$. It estimates the capabilities of the features selected to solve tasks independently. After each task, we freeze the weights of past heads to avoid forgetting. After training, we concatenate all the heads into a single linear classifier and evaluate this classifier on the whole scenario (the test performance with a single head is noted $A_{te-global\_softmax}$). This second evaluation estimates the generalization capabilities of the features selected.

Hence, if classifiers select local spurious features, $A_{te-local\_softmax}$ should be significantly bigger than $A_{te-global\_softmax}$.

## 5.2 DISENTANGLING FACTORS OF INFLUENCE

Apart from the problem of feature selection, a gap between $A_{te-local\_softmax}$ and $A_{te-global\_softmax}$ could be explained by (1) the difference in difficulty of both evaluation (multi-head and single head), (2) an unbalance of bias and norms from different heads that could lead to bad performance in *single*

*head* (Lesort et al., 2021) (more details about this problem in appendix G). However, if neither (1) or (2) are sufficient to explain the performance gap, then we can conclude that LSFs exists.

**(1) Difficulty gap:** the comparison between single head and multi-head is biased because the first is one 10-way classification (harder) while the latter is the addition of five binary classifications (simpler). To estimate the difference of difficulty between both, we added a non-parametric method, *MeanLayer* as in (Lesort et al., 2021) which is an nearest mean classifier (NMC). There is no feature selection in MeanLayer. The classifier only uses the mean of the features of each class. There is no feature selection then no problem with local features selection. The difference in performance with MeanLayer in multi-head and the single head is then a good proxy to estimate the difference in the difficulty of both evaluations.

**(2) Unbalance of bias and norms:** we compare the linear layer performance with the weightnorm layer from (Lesort et al., 2021). This layer does not use norm and bias for inference and is then insensitive to such imbalance (details in appendix G).

We note that forgetting is by design impossible here since the features extractor and the other heads are frozen. Hence, forgetting can not explain drops in performance.

### 5.3 LOCAL SPURIOUS FEATURES EXPERIMENTS

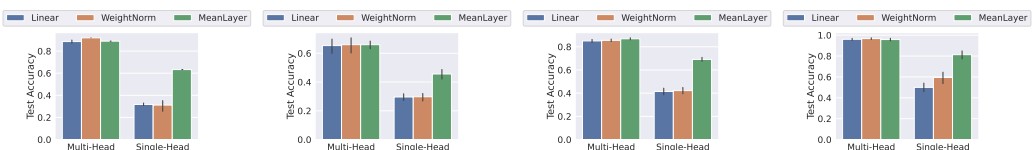

Figure 8: Comparison of final test accuracy $A_{T-1}$ in multi-head and the same classifier without task label information (In this order: CIFAR10, CUB200, OxfordFlower102, OxfordPet). The results are averaged among several pretrained models. The performance gap in the linear layer indicates how local features selected are generalizable. We use the weightnorm to estimate if bias or norm imbalance phenomena (Lesort et al., 2021) play a role in the performance gap . The MeanLayer baseline indicates the difference of difficulty of both evaluations. The results of this figure indicate that classifiers rely on local spurious features to solve tasks.

We report results in Figure 8. We remind that only the inference method changes between single head and multi-heads results. The parameters of the classifiers are the same (cf Sec. 5.1). The details of the results are in appendix H. We see that there is a considerable accuracy gap between multi-head (MH) $A_{te-local\_softmax}$ and single head (SH) $A_{te-global\_softmax}$ performance in the linear layer. Moreover, the gap for MeanLayer is significantly smaller than the gap for the linear layer. Hence, the difference in difficulty is not sufficient to explain the accuracy gap of the linear layer. And finally, the results are similar for linear layer and weightnorm which show that the norm/bias unbalance is not a problem that could explain the performance gap. Therefore, as described in Sec 5.1, we can conclude that *the classifier selected local spurious features, which leads to a poor generalization and a low single-head performance.*

These phenomena prove that the drop in performance in class-incremental can be due to a bad feature selection and not necessarily to forgetting. An explanation for this is that in CL, the unavailability of data from future tasks makes it impossible to know the features that generalize to the whole scenario. Therefore the model solves the tasks based on currently available data and can only verify later if the feature selected are correct. This is a fundamental observation for future continual learning approaches as it means that the features learned and selected should not necessarily be preserved for later tasks. While the continual learning bibliography focused mostly on forgetting or transfer, the LSF problem is also a great challenge to deal with to create efficient CL.

Another interesting insight that this experiment gives is that contrary to experiments in section 4.3.3 which show that using a *pre-trained model can help for spurious features, it does not offer a solution to local spurious features.* A good feature extractor is then not sufficient for a good feature selection.

## 5.4    Effect of Number of Classes per Tasks

We have seen in the previous experiments that local spurious features are due to wrong selection of features. Nevertheless, if we increase the complexity of a task, for example, by increasing the number of classes per task, then the model is forced to learn more about the data and select more features, building a more complete representation of the data. In this experiments, we investigate the influence of local spurious features with various number of classes per tasks. The hypothesis is that increasing the number of classes should reduce the influence of local spurious features. We create scenario of 5 tasks with various number of classes per tasks from tiny-Imagenet and CUB200 and train only a classifier on top of a pretrained model in a similar way as in previous experiments (Sec. 5.1).

Fig. 9 reports the difference $\Delta_{Head} = A_{te-local\_softmax} - A_{te-global\_softmax}$ for a various number of classes per task. For each scenario, one task represents 20% or the total number of classes. Hence, the difference of difficulty between solving tasks separately (MH) and together (SH) is assumed to be marginal. The results with MeanLayer in Fig. 9 confirm this hypothesis; the gap stays quite stable with number of classes per task. On the other hand, Fig. 9 shows that increasing the number of classes per task reduces $\Delta_{Head}$, which means that the influence of local spurious features is reduced. In practice, the algorithms should be more careful about local spurious features when the number of classes in a task is low.

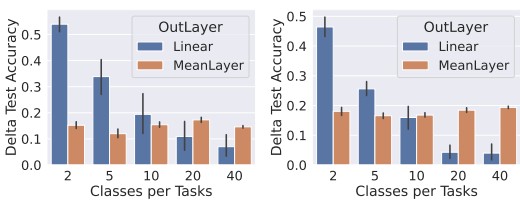

Figure 9: Delta accuracy between MH and SH accuracy. The effect of LSF reduces with the number of classes per tasks. CUB200 (left), TinyImagenet (right), 5 tasks per scenario.

**Conclusion:** In this section, we have shown that local spurious features (LSF) exist. We proved it by highlighting their influence on a classifier on top of frozen pretrained models. We consider that it is sufficient to assume that the same problem would also exist when training the models end-to-end. We also showed that the influence of LSF depends on the number of classes inside a task. The more classes there are inside a task, the fewer influence LSFs have.

## 6    Conclusion

Continual learning algorithms are built to learn, accumulate and memorize knowledge through time to reuse them later. Memorizing bad features can have catastrophic repercussions on future performance. Then, to learn general features, algorithms need to deal with spurious and local spurious features.

This paper first investigates the impact of spurious features on continual learning. Algorithms easily overfit spurious features for one or several tasks, leading to poor generalization. Our goal was to investigate if continual learning algorithms can benefit from the variation of the spurious features through time to ignore them. Our results show that a classical continual learning approach such as rehearsal can deal with spurious features until a certain level of correlation with labels. Secondly, we investigated if local spurious features exist in usual CL scenarios and can compromise continual learning. Those features are supposed to correlate spuriously with labels within a task only. Our results show that they are indeed present, and they can influence performance in CL significantly.

In the continual learning literature, performance decrease is generally attributed to catastrophic forgetting. Our results show that the problem of local spurious features also plays a major role. More research is needed to understand better the impact of local spurious features along with catastrophic forgetting. Understanding this phenomenon is critical to better address forgetting and feature selection and enable efficient continual learning.

We recommend using replay as a practical solution to reduce SPs and LSPs influence. Indeed, the replay process can simulate an identically and independent distribution of tasks seen so far. It makes it possible to fix potential mistakes in past feature selections. Nevertheless, even if this solution already exists, it significantly increases the computational cost when the number of tasks grows. Therefore, a better understanding of forgetting and spurious feature in CL could help to optimize it.

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

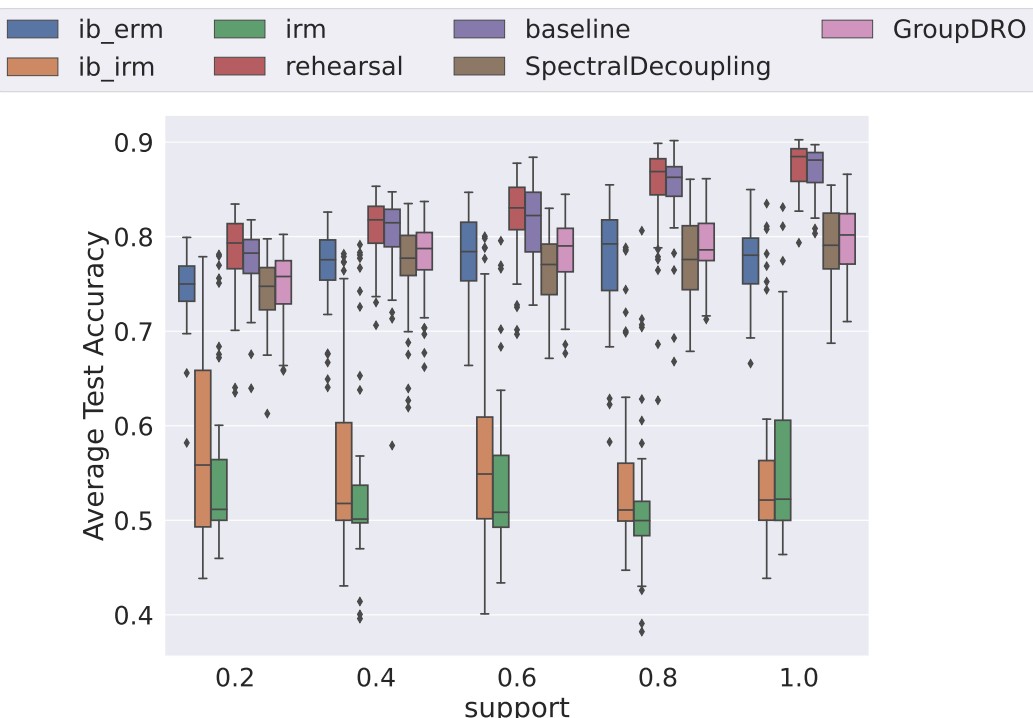

Figure 10: Averaged accuracy $\Omega$ on 10 tasks over various amount of support, the spurious correlation is set to $0.75$.

Shiori Sagawa, Pang Wei Koh, Tatsunori B Hashimoto, and Percy Liang. Distributionally robust neural networks. In *International Conference on Learning Representations*, 2019.

Bowen Zhao, Xi Xiao, Guojun Gan, Bin Zhang, and Shu-Tao Xia. Maintaining discrimination and fairness in class incremental learning. In *Proceedings of the IEEE/CVF Conference on Computer Vision and Pattern Recognition*, pp. 13208–13217, 2020.

Hattie Zhou, Ankit Vani, Hugo Larochelle, and Aaron Courville. Fortuitous forgetting in connectionist networks. In *International Conference on Learning Representations*, 2022. URL https://openreview.net/forum?id=ei3SY1_zYsE.

## A COMPUTE

The experiments were run on internal cluster on Quadro RTX 8000 GPUs. The total time of compute for an approximate period of 100 days of GPU use with hyper-parameters selections and experiments.

## B LOWERING THE SUPPORT OF THE DISTRIBUTION AT EACH TASK

### B.0.1 LOWERING THE SUPPORT AMOUNT

In experiments Sec. 4.3.2, the same data support was used for all tasks. This means that the shared support of original data is 100%. However, we expect continual algorithms to learn with less support. The support of a task is the percentage of the full original distribution within the task. We investigate in those experiments the influence that the support amount has on learning algorithms. We set the spurious correlation to 75% in those experiments.

As it is designed in the SpuriousCIFAR2 scenario, each class (0 and 1) has 5 distributional modes corresponding to the 5 original classes. In the previous experiments, all the data for all those modes are available in all tasks only the spurious features changed from one task to another.

In those experiments, we reduce the support by subsampling the original data. We select the data of a subset of the CIFAR10 classes for each task. For example, we select only cars for class 0 and only deers for class 1, instead of airplanes, cars, trucks, ships, and horses for class 0 and birds, cats, dogs, deers, and frogs for class 1. Hence, if we use the support of 0.2 (i.e. 20%), we will only select the data of $5 * 0.2 = 1.0$ original classes for class 0 and one other for class 1. For simplicity's sake, we will use only support compatible with the number of classes to have a round subset, i.e., $[0.2, 0.4, 0.6, 0.8, 1.0]$. At each task, the support is randomly sampled, hence the same original data can be in several tasks, but the spurious features will still be different for all tasks.

The support experiments results, illustrated in Fig. 10, show that in all support settings, the finetuning baseline and rehearsal are the best performing methods. On the other hand, contrary to what would be expected, the amount of support seems to not play an important role in the final performance, at least in the range of supports possible in our scenario. This is probably because doing replay converts a partially observable setting into a fully observable setting by simulating an iid distribution.

## C    SPURIOUS CORRELATIONS: DIFFERENT CASES

We have defined the different types of features in Sec. 3. We can now identify different cases among the spurious correlation between features and labels. Those different cases can either apply to SF or LSF.

**Data Observability:** *(1) Fully observable data* (Javed et al., 2020) The global features are always present and always observable. In this case, we can assume that features in data that do not last are spurious, and we can learn to ignore them. *(2) Partially observable data* The global features' presence is not invariant. In this case, features that do not last can either be spurious or good.

**Noisiness of Spurious Features:**    *(1) SFs are irrelevant for other classes* they can be ignored completely without affecting the learning process. We will refer to them as noisy SFs.    *(2) SFs are good features for other classes*, e.g., for classification, the color can be a spurious feature for some classes and valuable for others. We can not ignore those features since it could lead to poor performance in other classes.

In our experiment, we propose settings with fully observable and partially observable data. Our spurious features are noise in our domain incremental experiments and true features in our class-incremental experiments. Nevertheless, we do not exploit information about spurious feature types to design approaches.

## D    DISCUSSION

**Spurious Features vs Local Spurious Features:**  Spurious features and local spurious features lead to the same problem for learning algorithms: overfitting features that are not discriminative. The difference is that spurious features result from a covariate shift between training and test data. In contrast, local spurious features are due to the unavailability of all data while learning. Local spurious features are then, more specifically, a continual learning challenge, and we showed that they could have a significant impact in classical continual learning scenarios.

**Solutions to spurious features:**  The problems of spurious features and local spurious features lead to phenomena where forgetting is helpful to improve final performance (Zhou et al., 2022) also denoted as graceful forgetting. Indeed, forgetting by reinitializing some weights allows escaping spurious local minima and relearning a better solution taking new data and knowledge into account. On a more general note, spurious features and local spurious features make ineffective approaches that are too rigid and unable to modify and fix previously learned features/knowledge. The replay methods have been known to be a good solution for many continual learning problems. Our findings do not challenge this approach. However, it could be of some use to make replay approaches more effective and reduce the need for replay.

**Benchmark:**  The scenario SpuriousCIFAR2 proposed in this paper has been designed to highlight the problem that spurious correlation might create in continual learning. This scenario plainly fulfills its task of disturbing CL algorithms, particularly in the 100% correlation setting. However, it can not be used as a benchmark to evaluate the robustness of algorithms. The spurious features are very

simple, and a simple ad hoc processing of data could solve this scenario, i.e., encoding data with a pre-trained model as in Fig. 7a. A proper benchmark to assess robustness to spurious correlation would propose spurious features easy to learn by the model but harder to detect or ignore than simple squares of color. DomainBed datasets (Gulrajani & Lopez-Paz, 2020) are an interesting set of benchmarks for spurious correlation investigation. However, the amount of features is not controllable, making it harder to evaluate the limits of algorithms.

## E  SAMPLES SUPPORT EXPERIMENTS

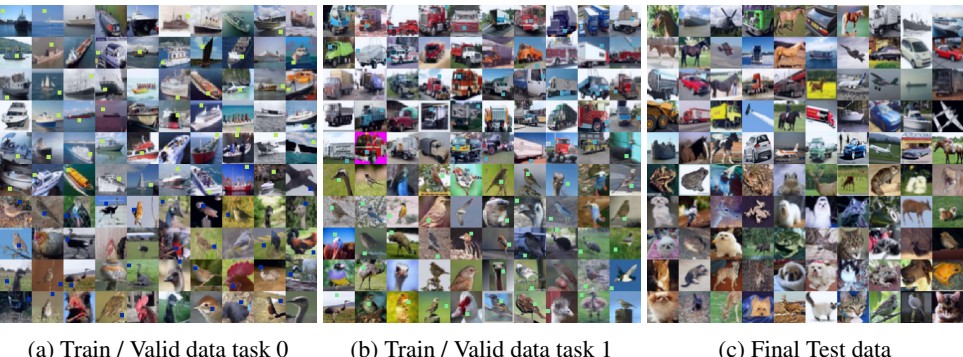



(a) Train / Valid data task 0      (b) Train / Valid data task 1      (c) Final Test data



Figure 11: Samples for support experiences, here with 20% support, i.e. the data of only two of the original classes in each task.

## F  HYPER-PARAMETERS SELECTION

For a fair comparison between algorithms originally designed for continual learning, such as replay, and OOD algorithms, we conduct the hyper-parameter search more intensively for OOD approaches.

For each OOD approach, we search for the best hyper-parameters in the range proposed in the DomainBed GitHub repository [1]. We also searched for the best learning rate with the bayesian method of wandb (Biewald, 2020). We used approximately 100 runs for each OOD baseline to select hyper-parameters. The scenario for hyper-parameters selection was an OOD setting with 5 environments of SpuriousCIFAR2 with 75% correlation but all simultaneously available (with no continual stream of tasks).

The hyper-parameters for rehearsal and finetuning (baseline), have been selected on a finetuning training in a single task setting with 75% of correlation. The number of samples per class for replay has been selected on a 5 task SpuriousCifar2 scenario with 75% of correlation. The optimizer used was Adam with a learning rate of 0.06 (others HPs are default ones in pytorch library).

## G  BIAS NORM IMBALANCE

As described in (Lesort et al., 2021): A linear layer is parameterized by a weight matrix $A$ and bias vector $b$, respectively of size $N \times h$ and $N$, where $h$ is the size of the latent vector (the activations of the penultimate layer) and $N$ is the number of classes. For $z$ a latent vector, the output layer computes the operation $o = Az + b$. We can formulate this operation for a single class $i$ with $\langle z, A_i \rangle + b_i = o_i$, where $\langle \cdot \rangle$ is the euclidean scalar product, $A_i$ is the $i$th row of the weight matrix viewed as a vector and $b_i$ is the corresponding scalar bias.

It can be rewritten:

$$\|z\|\|A_i\| \cdot cos(\angle(z, A_i)) + b_i = o_i \tag{3}$$

Where $\angle(\cdot, \cdot)$ is the angle between two vectors and $\|\cdot\|$ denotes here the euclidean norm of a vector.

---

[1] https://github.com/facebookresearch/DomainBed

Then, at inference time, $y_i = argmax_i(o_i)$ rely on the norm of $\|A_i\|$ and on the bias $b_i$. Within a single task, i.e. within a single head in a multi-head setting, $\|A_i\|$ and $b_i$ are balanced to predict class correctly. However, we can not ensure that $\|A_i\|$ and $b_i$ will are not biased from one head to another.

To avoid unbalance for bias and norm for inference, (Lesort et al., 2021) proposed the *weigthnorm* layer where: $\|z\| \cdot cos(\angle(z_t, A_i)) = o_i$ and show that this layer in efficient in learning in incremental and lifelong settings.

## H  DETAILS MULTI-HEAD EXPERIMENTS

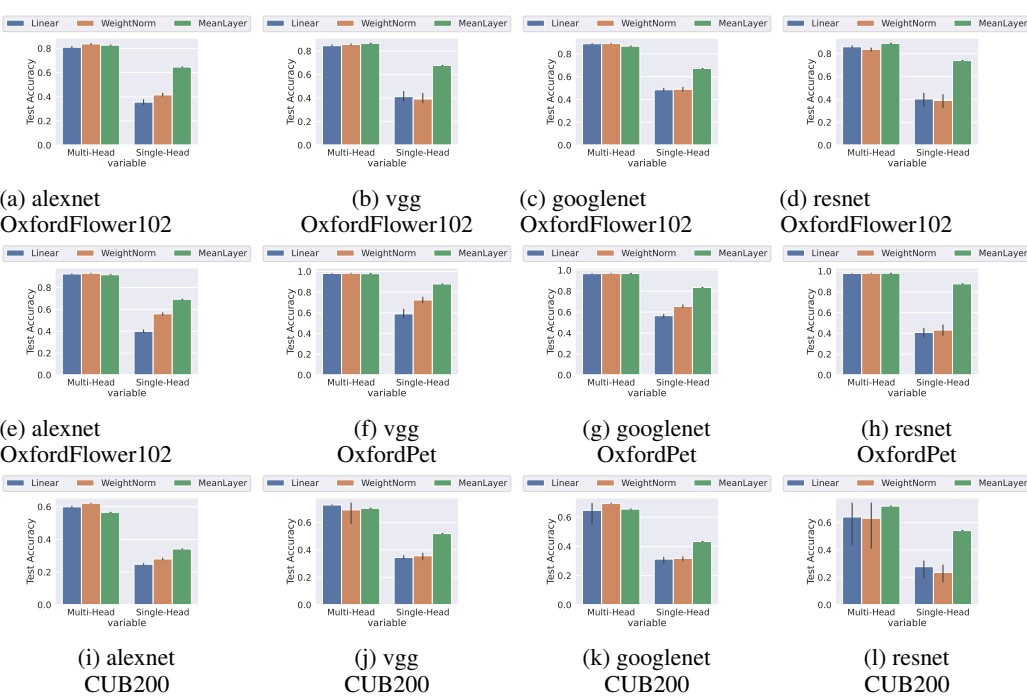

(a) alexnet
OxfordFlower102

(b) vgg
OxfordFlower102

(c) googlenet
OxfordFlower102

(d) resnet
OxfordFlower102

(e) alexnet
OxfordFlower102

(f) vgg
OxfordPet

(g) googlenet
OxfordPet

(h) resnet
OxfordPet

(i) alexnet
CUB200

(j) vgg
CUB200

(k) googlenet
CUB200

(l) resnet
CUB200

Figure 12: Local spurious features experiments: after training in a multi-head way, we compare accuracy between multi-head (soft-max applied on a subset of classes' outputs) and single head (softmax applied on all classes' outputs). The performance differences assess how the model selected local spurious features to solve tasks. Two baselines are added, (1) meanlayer, which assesses the difference in the difficulty of the two evaluations, and (2) weightnorm, which assesses if the performance difference is due to an imbalance in norm or bias. The results in this figure show that models indeed rely on local spurious features to solve tasks.

## I  SAMPLING ALGORITHM

Algorithm 1 describes the sampler used to train the model with replay. The input dataset $\mathcal{D}$ is a concatenation of the buffer with the current data. The goal is to make the probability of sampling on each class, the same whatever the number of sample for each class in $\mathcal{D}$.

**Algorithm 1** Balanced Sampling of Data Mixture.

1: **procedure** GET_SAMPLER($\mathcal{D}$)
2:     $y \leftarrow \mathcal{D}.y$                                                                          ▷ Get data class labels
3:     nb_per_class = bincount(y)                                                   ▷ count the number of occurence of each class
4:     $weights\_per\_class = \frac{1}{nb\_per\_class}$
5:     sample_weights = weights_per_class[y]                          ▷ give a sample probability weight to each data point
6:     sampler = Sampler($\mathcal{D}$, sample_weights, replacement=True)   ▷ create sampler to sample accordingly to the sample_weights
7:     **return** sampler
8: **end procedure**

