# OpenReview forum: "Spurious Features in Continual Learning"
_ICLR.cc/2023/Conference — Submitted to ICLR 2023_

### Official Review · Reviewer_mcta · 2022-10-24

**Confidence:** 4
**Correctness:** 3
**Technical Novelty And Significance:** 2
**Empirical Novelty And Significance:** 2
**Recommendation:** 5

**Clarity, Quality, Novelty And Reproducibility:**

Certain parts of the paper are really hard to follow. The experiments section is quite convoluted. Misplaced figures and sections, baselines that are either not described or described only after the results are reported. The paper should be restructured. Section 5 is particularly hard to understand. To the extent that is difficult to distill the key takeaways.

Code is not provided so results can't be reproduced

**Strength And Weaknesses:**

### Strengths

The paper does a nice job of classifying the different types of features you can find on CL tasks (spurious vs. good, and local vs. global). These are not only described with intuitive examples but also defined mathematically.

The experimental evaluation is thorough. The paper also provides valuable insights that might be useful for the community. Such as the fact that using a rich enough replay buffer is sufficient when the amount of spurious correlation is low enough; or the idea of using regularization in the opposite way (forcing the model to use more features). The paper also investigates simple tricks that help mitigate the problem of spurious correlations. Such as using a pre-trained model when possible, or using spectral decoupling to maximize the number of features selected.

### Weaknesses
The paper does not contribute any new solution to the problem of spurious correlations in CL. I can foresee that this is also going to be the main point of criticism from other reviewers. In my opinion, proper analysis of the problem and empirical evaluation of existing methods should be sufficient for acceptance. That being said, CL is not my area of expertise. Some of the insights that the paper brings forward were interesting and valuable for me but I am not sure to what extent these were already known by the community.

Given that the main contribution of this paper is the empirical evaluation of existing approaches and solutions to the problem of spurious correlations, I feel like the experiments section should be very clearly structured such that future researchers and practitioners can use it as a reference when facing similar types of problems.

Section 5 is very hard to follow. A new approach (multi-head), a set of experiments, and problems with the comparison (single head vs. multi-head, bias, and norm unbalance) are all described in the same place. I suggest breaking things down into multiple subsections such that the structure is more clear. There are also a lot of repetitions. My suggestion would be to first list what things you are investigating, then explain how these are investigating (maybe also explaining why the comparison is hard), then describe the results, and finally summarize the key points.

**Comments on Section 5.3:**

I'm having trouble understanding the difference between multi-head (MH) and single-head (SH). Where is this explained? The first reference to this is in Section 5.1:

*"We use a multi-head approach to see if the classifier learns local features. Hence, while training, we only apply the softmax function to current classes outputs. (...)  After each task, we freeze weights of past heads to avoid forgetting. After training, we compare the test performance with the same classifier (i.e., the classifier trained with multi-head) but with the softmax applied on all
outputs Ate−global softmax,"* What does this mean?

Do I understand correctly that in single-head you use the same layer for all tasks, while for multi-head you replace the layer at each task during training?

If I understand correctly, the takeaway from this experiment is that if you increase the number of classes that the NN needs to predict in each task then it is less prone to overfitting to spurious features and can perform better on the test task. I feel this is somewhat obvious. Am I missing something?

Figure 9: which one is single-head and which one is multi-head?

"For each scenario, one task represents 20% or the total number of classes" What does this mean?

**Minor details:**

Move Figure 5 before Figure 6.

I suggest moving Section 4.3.3 before the experiments. It is weird that the methods are described after the results are reported. Also, including a brief description of all the other baselines would be useful.



**Summary Of The Paper:**

This paper offers an extensive empirical evaluation of different methods for continual learning (CL). It shows that some methods are unable to find what the paper calls good features. That is features that generalize across tasks and also to the test set. The algorithms are evaluated on various datasets and under different amounts of spurious correlation between spurious features and labels. Insights into why some algorithms work better than others and under what conditions are also given. Moreover, the paper distinguishes a new type of correlation that the authors call local spurious correlation. The consequences of this on model performance are also investigated empirically.


**Summary Of The Review:**

I already reviewed this paper for NeurIPS. There have been no major changes to the paper aside from an additional section (5.3). Hence, I am copy-pasting my previous review and adding a few comments about Section 5.3. Overall, I think the paper is still not ready for publication. Although there are some interesting insights the paper does not have a clear structure which makes it hard to parse.

---

### Official Review · Reviewer_g5vD · 2022-10-24

**Confidence:** 5
**Correctness:** 2
**Technical Novelty And Significance:** 2
**Empirical Novelty And Significance:** 2
**Recommendation:** 3

**Clarity, Quality, Novelty And Reproducibility:**

The paper is easy to read.
It is an analysis paper, but unfortunately, I fail to see any new insights that aren't already known.
No code submission. But since the issues discussed in the paper are fairly obvious, the reproducibility is not an issue.


**Strength And Weaknesses:**

Strength
The paper experimentally demonstrates spurious features harm continual learning, in particular class-incremental learning. This is valid.
The writing is not bad.

Weaknesses
1. Your examples of spurious features seem confusing. For the general spurious features, you give the example, “… but in the training data, all cars are red, and all bikes are white, but in test data, both are in a unique blue not available in train data. A model could easily overfit the color to solve the task while it is not discriminative in the test data. In CL, we would expect future tasks to bring pictures of cars and bikes of other colors to learn better features.” When you present local spurious features, you said “because only red cars and white bikes are currently available in train data, but in the test data, there are also cars and bikes of colors not seen yet or seen in the past,” I fail to see the difference between the two examples for different types of spurious features.

2. The above examples have two problems:

(i). Spurious features apply to any supervised learning, not specific to continual learning. The problem has been investigated extensively in supervised learning and thus it has not been the focus of the continual learning except (maybe) in online continual learning.
(ii). In the continual learning context, you never know what classes will come in the future. The color features may not be the best for the current task but may be useful to some future tasks. Thus, calling the color features spurious is not justified as continual learning is different from traditional supervised learning. This problem has been discussed in [a] and many approaches have been also proposed to deal with it [a, b, c],

(a) Guo et al. Online Continual Learning through Mutual Information Maximization. ICML, 2022.
(b) Zhu et al. Class incremental learning via dual augmentation, NeurIPS, 2021.
(c) Zhu et al. Prototype Augmentation and Self-Supervision for Incremental Learning, CVPR-2021.
3. I don’t think that the types of features you analyzed in your examples should be called spurious features because they are not un-useful. They are just not enough. True spurious features should be ignored.

4. In section 4.3.3, you give two recommendations, (i) using pre-trained model and (ii) maximizing the amount of features selected. These have been done. For (i), please see the following two papers and for (ii), please see [a, b, c].

(d) Kim et al. A Multi-Head Model for Continual Learning Via Out-of-Distribution Replay, Collas’2022.
(e) Wang et al., Learning to prompt for continual learning. CVPR’2022
In fact, all continual learning papers in natural language processing use pre-trained models. In computer vision, papers are starting to use pre-trained models too.

5. The paper proposed a modified version of Out-of-Distribution (OOD) generalization methods for continual learning and evaluated using the SpuriousCIFAR2 data, but this has been done and successfully used in continual learning, please see [d, f].

(f) Kim, et al. Continual Learning Based on OOD Detection and Task Masking. CVPR workshop, 2022.
6. For the analysis of local spurious features, your setting is not a continual learning setting as you treat each task independently, which clearly has no forgetting, but that is not continual learning. Your pre-trained model has label and information leak because some of your datasets are closely related to or overlapping with the ImageNet data. In your multi-head setting, the local spurious features are obviously.

7. For the local spurious features, which are the main problem of CIL, you did not give any new recommendations to solve the problem except saying replay is a practical solution. That is true, but that is what replay methods are designed for. I don’t see any new insights.

8. Please state explicitly which problem you are dealing with, class-incremental learning (CIL) or task-incremental learning (TIL). I believe that you are dealing with CIL based on your problem formulation and later analysis.



**Summary Of The Paper:**

This paper is an analysis paper. It experimentally shows that spurious features and local spurious features harm continual learning.

**Summary Of The Review:**

This is purely an analysis paper, but I fail to see any new insights from the analysis. The main issue is that the authors are unaware of many existing continual learning papers, which have already discussed and dealt with the problems that they pointed out.

---

### Official Review · Reviewer_9EtC · 2022-10-26

**Confidence:** 3
**Correctness:** 2
**Technical Novelty And Significance:** 1
**Empirical Novelty And Significance:** 2
**Recommendation:** 3

**Clarity, Quality, Novelty And Reproducibility:**

The clarity of the paper is poor in general (see above). The empirical evaluation is novel to the best of my knowledge, but the paper does not have any methodological novelty.

**Strength And Weaknesses:**

Strengths:
- The paper studies the intersection of two important machine learning problems (spurious correlations + continual learning), which has not been examined in prior work to the best of my knowledge.


Weaknesses:
1. The paper misses a large set of recent works in the spurious correlation literature, including but not limited to [1-5]. Many of these works should be included as benchmark methods, and the result in [4] specifically might be useful to explain some of the findings in Section 5.

2. The experimental setting in the paper is rather limited. The experiment in Section 4 is only conducted on a single semi-synthetic dataset, which is not one of the common spurious correlation datasets from the literature. The authors should expand this experiment to standard datasets like Waterbirds and CelebA, where the spurious correlation can vary across tasks by supersampling or by introducing it semi-synthetically.

3. The experiment in Section 5.1 is confusing to me. As the authors do not know the level of spurious correlations in base CIFAR10 or CUB200, it is unclear to me how the authors are disentangling the effect of the local spurious correlation. The authors should consider exploring the effect of introducing semi-synthetic spurious correlations in these datasets. The authors should also evaluate mitigating algorithms in this setting.

4. The clarity of the paper is poor in general. In particular, the setup of the experiment in Section 5.1 is hard to understand, and the formulation of spurious features in Section 3 can be improved. For example, $C_T$ should be defined formally along with the correlation function, and the definition of $z_+$ (Eq. 1) seems restrictive as it does not take non-linear feature interactions into account. The authors should consider defining their problem setting more formally, such as the formulation in [6].

5. There are many spelling and grammar mistakes throughout the paper. The language used in the paper should also be made more formal in general. For example, "good feature" should be replaced with "invariant feature". The "test data" is also overloaded (as is clear in the first paragraph of Section 4.3.1). The authors should also clearly state which methods require knowledge of the spurious attribute, and which do not.

6. The scope of paper is quite limited. The authors evaluate existing methods in a couple of scenarios, but do not provide any novel methods for addressing the problem.

[1] https://arxiv.org/abs/2107.09044

[2] https://openreview.net/forum?id=jphnJNOwe36

[3] https://proceedings.neurips.cc/paper/2020/file/f1298750ed09618717f9c10ea8d1d3b0-Paper.pdf

[4] https://arxiv.org/abs/2204.02937

[5] https://arxiv.org/abs/2201.00299

[6] https://arxiv.org/abs/2010.15775

**Summary Of The Paper:**

The authors study the problem of spurious correlations during continual learning. They divide the problem into two cases: the typical spurious correlation setting, and a local spurious correlation setting where the spuriousness is due to limited data within each task. They conduct experiments for each of the two cases, and compare with baselines from domain generalization. They conclude that spurious correlations may lead to, and may be responsible for, large performance drops during continual learning.

**Summary Of The Review:**

The paper examines an interesting and previously unstudied problem. However, the scope of the paper is rather limited (Weakness #6), and there are major issues with the empirical evaluation (Weaknesses #1-3), as well as with the clarity of the paper (Weaknesses #4-5). For these reasons, I recommend rejection.

---

### Official Review · Reviewer_22hc · 2022-10-27

**Confidence:** 4
**Correctness:** 2
**Technical Novelty And Significance:** 2
**Empirical Novelty And Significance:** 2
**Recommendation:** 3

**Clarity, Quality, Novelty And Reproducibility:**

Please see the comments above. Overall, there are some missing explanations on setting, justification, etc. So, I think the clarity/quality is not very high.

**Strength And Weaknesses:**

Strength:
- The paper proposes to consider new notion of LSF in continual learning problems.
- The paper attempts to produce various data settings to validate their claim.

Weakness:
- First of all, since this is an analysis paper, I think the experimental setting for each figure should have been more clearly presented.
   - For example, in each figure, are test accuracies all obtained from the final test set?
   - How are tasks formed in the continual learning experiments?
   - When the average accuracy is computed, how did you select the hyperparameters? (based on the final acc? this may not make sense in the continual learning setting.)
- It is not entirely clear whether the SF and LSF really exist in practice. When exactly does this features exist? Can you characterize them?
- Setting for the SF experiments
   - Only the setting with same correlation level across tasks is considered. What about when the correlation levels are different across tasks? What happens then?
- Trivial results in Table 5
   - When the spurious correlation is low, the baselines without bias remedy perform well, while when the correlation is high, group dro is performing well. This might be a common result even for the single task, and what information does this result additionally give for the continual learning setting?
- Unclear definition of LSF?
   - LSF is defined as features that can be useful for a specific task, but are spurious for *all* tasks. But, this definition would also classify the useful features as LSF as well. For example, when a feature is useful for a certain task of classifying cat from dog, it could become LSF when the all tasks involves two breeds of dogs. Is it really LSF?
- LSF results are unclear
   - The paper considers Task-IL setting and compares the performance between single-head and multi-head structures. The conclusion is that since Multi-head structure has high accuracy, LSF is causing a problem in single head. But it seems like there is some logical gap in this result -- multi-head might be using more information to achieve high acc, but there is not consideration for this aspect.

**Summary Of The Paper:**

The paper mainly studies about the spurious features (SF) and local spurious features (LSF) in continual learning. It is an analyses paper without a method to address the observed problem. The paper considers multiple settings involving SF and LSF and claims that the LSF problem is the main cause for the performance degradation in continual learning.

**Summary Of The Review:**

Please see the comments above. Overall, I do not think the paper meets the high bar of ICLR since several details and justification are missing.

---

### Decision · Program_Chairs · 2023-01-20

**Decision:**

Reject

**Justification For Why Not Higher Score:**

See above for weaknesses

**Justification For Why Not Lower Score:**

I recommend rejection

**Metareview: Summary, Strengths And Weaknesses:**

The paper mainly studies about the spurious features (SF) and local spurious features (LSF) in continual learning. Although reviewers thought the topic is an important one, there are several weaknesses identified in the submission: the experimental setting for each figure is not very clear; The paper misses a large set of recent works in the spurious correlation literature; The experimental setting in the paper is rather limited; among others. I suggest authors to take a close look at the comments by reviewers to modify their paper accordingly.